# Boost 3D Reconstruction using Diffusion-based Monocular Camera Calibration

## Abstract

In this paper, we present *DM-Calib*, a diffusion-based approach for estimating pinhole camera intrinsic parameters from a single input image. Monocular camera calibration is essential for many 3D vision tasks. However, most existing methods depend on handcrafted assumptions or are constrained by limited training data, resulting in poor generalization across diverse real-world images. Recent advancements in stable diffusion models, trained on massive data, have shown the ability to generate high-quality images with varied characteristics. Emerging evidence indicates that these models implicitly capture the relationship between camera focal length and image content. Building on this insight, we explore how to leverage the powerful priors of diffusion models for monocular pinhole camera calibration. Specifically, we introduce a new image-based representation, termed Camera Image, which losslessly encodes the numerical camera intrinsics and integrates seamlessly with the diffusion framework. Using this representation, we reformulate the problem of estimating camera intrinsics as the generation of a dense Camera Image conditioned on an input image. By fine-tuning a stable diffusion model to generate a Camera Image from a single RGB input, we can extract camera intrinsics via a RANSAC operation. We further demonstrate that our monocular calibration method enhances performance across various 3D tasks, including zero-shot metric depth estimation, 3D metrology, pose estimation and sparse-view reconstruction. Extensive experiments on multiple public datasets show that our approach significantly outperforms baselines and provides broad benefits to 3D vision tasks.

## 1 Introduction

Camera calibration is a foundational task in 3D computer vision, critical for numerous applications such as camera pose estimation (Schönberger & Frahm, 2016), 3D reconstruction (Seitz et al., 2006), and zero-shot metric depth estimation (Yin et al., 2023). Traditional methods primarily focus on multi-view calibration, typically involving multiple images of fixed intrinsics (Pollefeys & Gool, 1997) or multiple images of checkerboard patterns (Zhang, 2000). However, these methods depend heavily on dense multi-view images with sufficient overlap, making them cumbersome and often impractical for sparse-view or even monocular setups. Consequently, monocular camera calibration has garnered significant research interest.

Monocular camera calibration is inherently an ill-posed problem, requiring additional information to address it. Traditional approaches have attempted to incorporate handcrafted knowledge, such as the gravity direction (Veicht et al., 2024), Manhattan World constraints (Liu & Cui, 2023), and human face priors (Hu et al., 2023). However, these handcrafted insights often fail to generalize effectively across diverse real-world scenarios. To overcome these limitations, recent studies (Zhu et al., 2023) recast monocular camera calibration as a learning-based regression problem, leveraging a single image to directly infer its intrinsic parameters.

While learning-based methods benefit from data-driven knowledge, outperforming traditional approaches, they are constrained by the limited availability of public datasets. As a result, these methods tend to overfit on training data and exhibit poor generalization to unseen scenarios. This limitation raises a critical question: *What kind of knowledge is necessary to develop a robust camera calibration method that exhibits strong generalization capabilities?*

One promising solution lies in leveraging stable diffusion priors (Rombach et al., 2022). The intuition behind stems from a key observation: stable diffusion models possess an implicit understanding of imaging across different focal lengths. As is widely known, *Cameras with long focal lengths tend to compress spatial relationships, resulting in a more flattened image perspective, while wide-angle cameras exaggerate depth and distance, resulting in a more pronounced perspective effect.* As illustrated in Fig. 1, we present two portrait images generated by a stable diffusion model (Rombach et al., 2022) using similar text prompts but with varying focal length descriptions. Clearly, the left image prompted with "long focal length",

**Text prompt:** *"A portrait of a woman standing by the window and looking down at the yard with a loving gaze, in a cinematic style. She is wearing an orange sweater, has short shoulder-length hair, and is holding a glass of water. Refer to the image of her. High quality image, long/short focal length."*

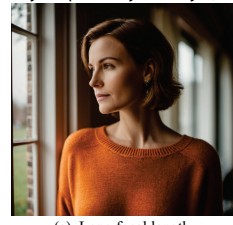 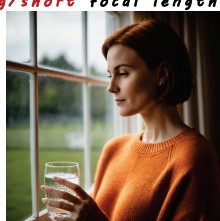

(a). Long focal length     (b). Short focal length

Figure 1: Images generated using text prompts that specify different focal lengths.

exhibits a more blurred background and shallower depth of field compared to the right image. This demonstrates that, by training on large-scale image-text pairs, these models encapsulate knowledge related to imaging characteristics associated with different focal lengths.

Despite these advancements, a key challenge persists: how to effectively leverage diffusion priors for high-precision camera calibration? In this paper, we set out to explore this question and introduce **DM-Calib**, a diffusion-based model for estimating intrinsic camera parameters from a single image. We recognize that the representation format used to encode camera intrinsics is essential for effective monocular camera calibration using diffusion models. To address this, we conduct in-depth investigation for various camera representations and develop the Camera Image, a novel image-based representation specifically engineered for seamlessly integration with pre-trained diffusion models, thereby preventing loss of information. Subsequently, we train a diffusion model that takes a single image as input and generates the Camera Image, followed by a RANSAC algorithm to solve the camera intrinsic parameters. Moreover, we demonstrate how to integrate the proposed camera calibration with diffusion-based metric depth estimation, which allows the recovery of true-scale depth measurements from a single image. Furthermore, our experiments show that the recovered camera calibration results significantly improve the performance of various downstream tasks, including camera pose estimation, sparse-view 3D reconstruction, and novel view synthesis, showcasing the robustness and effectiveness of *DM-Calib* in advancing accurate monouclar camera calibration.

To summarize, our main contributions are:

- We introduce the Camera Image, a novel image-based representation specifically designed to encode camera intrinsic information, optimized to use with pretrained diffusion models.

- We present **DM-Calib**, a generative foundation model that provides highly accurate estimations of camera intrinsics. Additionally, it can seamlessly integrate with various downstream tasks, showcasing its effectiveness and robustness to images from various scenarios.

- Extensive experiments on multiple public datasets show that our approach significantly outperforms baselines and provides broad benefits to 3D vision tasks.

## 2 RELATED WORK

### 2.1 MONOCULAR CAMERA CALIBRATION

Calibrating camera intrinsics, also known as self-calibration, is one of the most fundamental problems in geometric computer vision. Geometric methods typically assume multiple input images with the same intrinsic matrix. These methods can be broadly classified as direct methods and stratified methods. Direct methods (Zeller & Faugeras, 1996; Hartley, 1997; Luong & Faugeras, 1997) solve the Kruppa's equation (Gallego et al., 2018) to obtain camera intrinsic parameters, which are generally more fragile to noises. In comparison, stratified methods (Hartley, 1993; Triggs, 1997; Pollefeys & Gool, 1997) estimate camera intrinsics from a projective reconstruction by gradually recovering the affine and Euclidean structures such as the plane at infinity and absolute quadrics.

Traditional monocular camera calibration methods typically assume specific geometric structures to estimate intrinsics. For example, with Manhattan World assumption (Coughlan & Yuille, 1999), camera intrinsics can be inferred from vanishing points (Lee et al., 2013; Schindler et al., 2004;

Wildenauer & Hanbury, 2012), or by jointly estimating the horizon line (Zhai et al., 2016; Simon et al., 2018). Other approaches rely on calibration objects such as checkerboards (Zhang, 2000), line segments (Zhang et al., 2016), spheres (Zhang et al., 2007), pyramid frustums (Jiang et al., 2009), or even human faces (Hu et al., 2023; Liu & Cui, 2023). Despite producing satisfactory results, these methods are constrained by their reliance on specific objects, which limits their applicability in unconstrained, real-world scenarios. Recently, Zhu et al. (2023) introduced incident maps to regress camera intrinsics, enabling the detection of geometric manipulations like cropping.

Apart from the above geometry-based methods, some works attempt to leverage the strong generative models for camera calibration. To the best of our knowledge, the approach by He et al. (2024) is the only existing method that formulates camera intrinsic estimation as a generative task, leveraging incident maps with diffusion models. Although this method outperforms non-diffusion-based methods such as Zhu et al. (2023) in unconstrained settings, we argue that the full potential of diffusion models remains under-utilized and that its performance hinges on joint training with depth images. In contrast, our approach introduces a camera representation that is more inherently compatible with diffusion models, thereby eliminating the need to generate non-textured incident maps or rely on the joint training of additional geometric information.

## 2.2 DIFFUSION MODELS IN 3D TASKS

Recently, diffusion models (Ho et al., 2020; Song et al., 2020) have emerged as powerful tools across various domains, particularly in the field of computer vision, where Text-to-Image diffusion models (Saharia et al., 2022b) and their extensions (Poole et al., 2022; Rombach et al., 2022; Saharia et al., 2022a; Zhang et al., 2023) have garnered significant attention. Compared to GAN-based approaches (Bhattad et al., 2024), several studies have highlighted the advantages of diffusion models, especially when used as prior geometric cues in 3D tasks. Notable examples include view synthesis (Liu et al., 2023a; Long et al., 2024), camera calibration (He et al., 2024), normal estimation (Ye et al., 2024; Liu et al., 2023b), and depth estimation (Fu et al., 2024; Ke et al., 2024). In this work, we focus on leveraging diffusion models for camera intrinsic estimation, which serves as a foundation for enhancing a series of downstream tasks, such as monocular metric depth estimation, pose estimation, and sparse-view reconstruction.

## 2.3 MONOCULAR DEPTH ESTIMATION

For depth estimation, several works (Ke et al., 2024; Fu et al., 2024; Hu et al., 2024b) have shown that diffusion models can be fine-tuned to predict affine-invariant depth, achieving not only finer detail but also more accurate estimates than traditional methods (Ranftl et al., 2020; Yin et al., 2021; Yang et al., 2024; Yin et al., 2022). Recent studies (Xu et al., 2024; Ye et al., 2024; Garcia et al., 2024) have highlighted that diffusion models can serve as pre-trained networks for deterministic, one-step inference. However, none of the current approaches leverage pre-trained diffusion models for metric depth prediction. Existing zero-shot monocular metric depth estimation methods, such as (Bhat et al., 2023; Yin et al., 2023; Piccinelli et al., 2024; Hu et al., 2024a), have demonstrated accurate results, yet they still face challenges in capturing geometric details and foreground-background relationships, particularly in outdoor environments. Moreover, these methods usually employ contrastive learning pretrained encoders (i.e., DINO (Caron et al., 2021)) or classification pretrained encoders (Deng et al., 2009), and randomly initialize the decoder, which are trained on fewer images compared to diffusion models. Building upon this, we extend diffusion models to metric depth estimation. To the best of our knowledge, our approach is the first to employ pre-trained diffusion models for metric depth estimation, achieving finer geometric detail and competitive performance across diverse benchmarks.

## 3 METHOD

Given a single input image $\mathbf{x} \in \mathbb{R}^{H \times W \times 3}$, our objective is to recover its camera intrinsic matrix $\boldsymbol{K}$. To efficiently and losslessly integrate camera intrinsics prediction with diffusion models (Rombach et al., 2022), we introduce Camera Image (Fig. 3) to encode camera intrinsics as a detail-preserving color image (see Sec. 3.2). We reformulate camera calibration as a conditional generation task, transforming the text-to-image (T2I) diffusion model into an image-to-Camera-Image (I2C) model, from which camera intrinsics are recovered via a RANSAC algorithm (see Sec. 3.3). Our proposed calibration method significantly boosts the performance of downstream 3D vision tasks, including

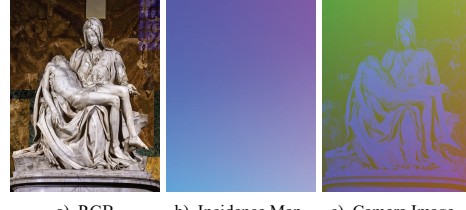

Figure 2: **Error analysis of camera representations.** We first use pre-trained VAE to encode and decode each camera representation, and plot the FoV reconstruction errors ($°$) here.

a). RGB      b). Incidence Map      c). Camera Image

Figure 3: **Visualization of incidence map and Camera Image.** We show the input RGB image, the incidence map and our proposed Camera Image for reference.

monocular metric depth estimation, 3D reconstruction, and pose estimation (see Sec. 3.4). Before introducing our method, we first revisit the preliminary concepts related to diffusion models.

## 3.1 PRELIMINARIES ON DIFFUSION MODEL

Diffusion models (DMs) (Ho et al., 2020) learn to model a data distribution $p_{\text{data}}(\mathbf{x})$ by progressively denoising a noise variable that is initially sampled from a normal distribution. Recognizing the efficiency issue associated with generating high-resolution images, Rombach et al. (2022) introduced latent diffusion models (LDMs), which operate the diffusion process in the latent space of a pretrained variational autoencoder (VAE) (Kingma, 2013) with an encoder $\mathcal{E}$ and a decoder $\mathcal{D}$.

For any given input image $\mathbf{x}$, the corresponding latent code is generated by the VAE encoder: $\mathbf{z} = \mathcal{E}(\mathbf{x})$. The forward diffusion process incrementally adds noise to these latents following $\mathbf{z}_t := \alpha_t \mathbf{z} + \sigma_t \boldsymbol{\epsilon}$, where $\boldsymbol{\epsilon} \sim \mathcal{N}(\mathbf{0}, \boldsymbol{I})$, and $\alpha_t$ and $\sigma_t$ are parameters defined by the noise schedule, with $t \sim p_t$ representing the time step within the diffusion schedule. The denoising network, denoted as $\boldsymbol{\epsilon}_\theta$, aims to reverse the diffusion process to recover the noise-free latent code $\hat{\mathbf{z}}$ from the final noisy latent code $\mathbf{z}_T$. This is achieved by predicting the noise component $\boldsymbol{\epsilon}_\theta(\mathbf{z}_t, t)$ at each diffusion step. The original image $\mathbf{x}$ is then reconstructed from this denoised latent code using the VAE decoder as $\hat{\mathbf{x}} = \mathcal{D}(\hat{\mathbf{z}})$. The whole diffusion model is optimized by minimizing the denoising score matching objective, defined as follows: $\mathbb{E}_{\mathbf{z}, \boldsymbol{\epsilon}, t} \left[ \|\boldsymbol{\epsilon} - \boldsymbol{\epsilon}_\theta(\mathbf{z}_t, t)\|_2^2 \right]$. This objective measures the squared Euclidean distance between the actual noise $\boldsymbol{\epsilon}$ and the predicted noise. By minimizing this objective, the denoising network learns to accurately estimate the noise, thereby effectively reversing the diffusion process and reconstructing the original data distribution.

## 3.2 CAMERA IMAGE REPRESENTATION

Monocular camera calibration aims to recover the camera intrinsics matrix $\boldsymbol{K}$, which typically composed of four parameters: $f_x$, $f_y$, $c_x$, and $c_y$, corresponding to the focal lengths and the optical center coordinates along the $x$-axis and $y$-axis, respectively. This numeric-based representation, however, does not align well with image-based diffusion models, which are primarily designed for generating spatial images. The challenge, therefore, becomes how to effectively leverage powerful pretrained SD models to retrieve implicit camera information.

To address this challenge, we propose a novel image-based representation, called "Camera Image", which encodes the camera intrinsic parameters into a 3-channel color image (refer to Fig. 3 for visual representation). This representation seamless integrates with existing diffusion models with minimal architecture modifications. We reformulate the camera intrinsics into a 2-channel pseudo-spherical representation defined by azimuth $\boldsymbol{\theta}$ and elevation $\boldsymbol{\varphi}$. This two-channel formulation enables us to explore the choice of the third channel to prevent mode collapse. Given that VAE encoders typically take a three-channel image as input, it is crucial to determine how to effectively fill the third channel. Simply duplicating one of the existing channels $\boldsymbol{\theta}, \boldsymbol{\varphi}$ or adding a constant value channel leads to suboptimal results, as detailed in Fig. 2. To enhance the camera representation, we propose a simple yet effective solution by incorporating the grayscale image $\mathbf{g}$ of the input $\mathbf{x}$ into the dense camera representation, reducing the domain gap between the input images and those generated by diffusion models. Consequently, our proposed camera image $\mathbf{c}$ is defined as follows,

$$\mathbf{c}_{(u,v)} = \left[ \arctan\left(\frac{r_1}{r_3}\right), \arccos\left(r_2\right), \mathbf{g}_{(u,v)} \right],  \tag{1}$$

Figure 4: **The overview training framework of *DM-Calib*.** The input image $\mathbf{x}$ and the camera image $\mathbf{c}$ are first encoded into latent space using a frozen VAE encoder. We then inject timestamp-relevant noise $\boldsymbol{\epsilon}$ into the camera's latent code, which is concatenated with the image latent code and fed into the subsequent UNet. The UNet is fine-tuned to predict the added noise $\hat{\boldsymbol{\epsilon}}$.

where $\vec{r} = [r_1, r_2, r_3] \cong \boldsymbol{K}^{-1}[u, v, 1]^T$, $\boldsymbol{K}$ is the intrinsic matrix, and $(u, v)$ are the pixel coordinates. $\vec{r}$ is normalized as a unit vector, and $\mathbf{g}_{(u,v)}$ is the gray-scale pixel value sampled at coordinate $(u, v)$. As shown on the right side of Fig. 3, the proposed camera image preserves the high-frequency details of the original scene, making it closely resemble real-world images that diffusion models are designed to process. The incidence map (shown in the middle of Fig. 3), proposed by concurrent research (He et al., 2024), however, exhibits a large domain gap from the original image domain, which results in suboptimal intrinsics estimations according to our experimental results in Sec. 4.2.

### 3.3 CAMERA INTRINSIC ESTIMATION

**Diffusion Model for Camera Image Prediction.** Our camera intrinsic estimation is built upon the pre-trained latent diffusion model, Stable Diffusion v2.1 (Rombach et al., 2022), which leverages robust image priors trained on the billion-scale LAION-5B dataset (Schuhmann et al., 2022). To facilitate the generation of the proposed camera image, we detail our training pipeline in Fig. 4. First, a frozen VAE $\mathcal{E}$ encodes the RGB image $\mathbf{x}$ and its corresponding camera image $\mathbf{c}$ into latent space $\mathbf{z}^{\boldsymbol{x}}$ and $\mathbf{z}^{\boldsymbol{c}}$, respectively. Multi-resolution noise (Kasiopy, 2023) $\boldsymbol{\epsilon}^{\boldsymbol{c}}$ is then added to the camera latents $\mathbf{z}^{\boldsymbol{c}}$, forming the noisy code $\mathbf{z}_T^{\boldsymbol{c}}$. This code is concatenated with $\mathbf{z}^{\boldsymbol{x}}$, serving as the input for the pretrained U-Net. To accommodate our inputs, we double the input channels of the original U-Net and adjust the corresponding parameter weights accordingly. The U-Net is targeted to predict the added noise, and the final loss function is expressed as:

$$\mathcal{L} = \mathbb{E}_{\mathbf{x},\mathbf{c} \sim p_{\text{data}}, t \sim p_t, \boldsymbol{\epsilon}^{\boldsymbol{c}}} \left\| \hat{\boldsymbol{\epsilon}}_{\boldsymbol{\theta}}(\mathbf{z}_t^{\boldsymbol{c}}; \mathbf{z}^{\boldsymbol{x}}) - \mathbf{v}_t \right\|_2^2, \tag{2}$$

where $\mathbf{v}_t = \alpha_t \boldsymbol{\epsilon}_t^{\boldsymbol{c}} - \beta_t \mathbf{z}_t^{\boldsymbol{c}}$, and $\boldsymbol{\epsilon}_t^{\boldsymbol{c}}$ is the sampled multiscale noise for the camera image. During inference, we can formulate the generation of the camera image within a generative framework $f: \boldsymbol{x} \in \mathbb{R}^3 \to \hat{\boldsymbol{c}} \in \mathbb{R}^3$ utilizing v-prediction (Salimans & Ho, 2022), as follows:

$$f(\mathbf{z}^{\boldsymbol{x}}) = p(\hat{\mathbf{z}}_T^{\boldsymbol{c}}) \prod_{t=1}^{T} p_{\boldsymbol{\theta}} \left( \hat{\mathbf{z}}_{t-1}^{\boldsymbol{c}} | \hat{\mathbf{z}}_t^{\boldsymbol{c}} \right), \tag{3}$$

where $\mathbf{z}$ is the latent feature and $\hat{\mathbf{z}}_T^{\boldsymbol{c}} \sim \mathcal{N}(\mathbf{0}, \boldsymbol{I})$. After completing the multi-step denoising process using the U-Net, the denoised camera latent representation $\hat{\mathbf{z}}^{\boldsymbol{c}}$ is sent to the frozen VAE decoder, yielding the final camera image $\hat{\mathbf{c}} = \mathcal{D}(\hat{\mathbf{z}}^{\boldsymbol{c}})$, and we verified that our camera image provides negligible error with respect to the VAE encoder-decoder reconstruction (see Fig. 2). From this generated image, we can extract the numerical representation of the camera intrinsic parameters.

**Recover Camera Intrinsics From Camera Image.** With the recoverd camera image $\hat{\mathbf{c}}$, camera intrinsic matrix $\boldsymbol{K}$ can then be inferred from the first two channels of the camera image via the relation between camera image and camera intrinsic $\boldsymbol{K}$ in Eq. 1 :

$$\tan(c_\theta) f_x + c_x = u, \qquad \frac{1}{\cos(c_\theta)\tan(c_\varphi)} f_y + c_y = v, \tag{4}$$

where $\hat{\mathbf{c}}_{(u,v)} = [c_\theta, c_\varphi, g]$ presents the pixel value of the camera image. Since, every two pixels can be used to solve the camera intrinsics, we employ the RANSAC algorithm to align the two lines using all pixel in the camera image. Here, the focal length $f_{x/y}$ and the optical center $c_{x/y}$ are represented as the slope and intercept of the best-fit line of Eq. 4, respectively.

Figure 5: **The overview of metric depth training pipeline.** The encoded image and camera image $z^x$ and $z_c$ are concatenated and sent to pretrained U-Net. Then we employ single-step diffusion at timestamp $T$ to generate depth latent code $\hat{z}_d$, which is then decoded into predicted metric depth $\hat{d}$.

Table 1: **Monocular Camera Calibration on Zero-Shot Datasets.** We report the calibration errors for both focal length and optical center. †: focuses on focal length prediction. ‡: Waymo and ScanNet are in the training set. ☀: joint training with depth.

| Method | Waymo | | RGBD | | ScanNet | | MVS | | Scenes11 | | Average | |
|---|---|---|---|---|---|---|---|---|---|---|---|---|
| | $e_f$ | $e_b$ | $e_f$ | $e_b$ | $e_f$ | $e_b$ | $e_f$ | $e_b$ | $e_f$ | $e_b$ | $e_f$ | $e_b$ |
| Perspective † | 0.444 | - | 0.166 | - | 0.189 | - | 0.185 | - | 0.211 | - | 0.239 | - |
| GeoCalib † | 0.285 | - | 0.203 | - | 0.137 | - | 0.104 | - | 0.344 | - | 0.215 | - |
| WildCame | 0.210 | 0.053 | 0.097 | 0.039 | 0.128 | 0.041 | 0.170 | 0.028 | 0.170 | 0.044 | 0.155 | 0.041 |
| DiffCalib | 0.188 | 0.053 | 0.092 | 0.018 | 0.089 | 0.041 | 0.135 | 0.032 | 0.108 | 0.029 | 0.122 | 0.030 |
| DiffCalib-D ☀ | 0.145 | 0.053 | 0.084 | 0.040 | **0.055** | 0.036 | 0.108 | 0.036 | 0.176 | 0.038 | 0.095 | 0.041 |
| Unidepth ‡ | - | - | 0.055 | 0.052 | - | - | 0.482 | **0.001** | 0.510 | 0.051 | 0.350 | 0.030 |
| Ours | **0.115** | **0.036** | **0.041** | **0.010** | 0.089 | **0.024** | **0.087** | 0.008 | **0.061** | **0.010** | **0.078** | **0.017** |

With the proposed camera image and intrinsics estimation, our approach offers applicability to various 3D downstream tasks, including monocular metric depth estimation (MMDE), camera pose estimation, and 3D reconstruction.

## 3.4 DOWNSTREAM 3D VISION TASKS

**Monocular Metric Depth Estimation.** To predict metric depth from a single image, the model must possess a deep understanding of the image perspective and estimate accurate intrinsic parameters of the camera. By leveraging the proposed camera calibration method, we repurpose diffusion-based image generators for accurate metric depth estimation. Previous works (Ke et al., 2024; Fu et al., 2024) mainly investigate affine-invariant depth estimation. However, we find the VAE decoder $\mathcal{D}$ can only predict values in limited range, thus limiting the performance of metric depth estimation. To fix this issue, we formulate stochastic multi-step denoise SD model as one-step deterministic forward process as shown in Fig. 5. Specifically, we first encode RGB image and our designed camera image $\hat{c}$ via VAE encoder into latent space, noting that no noise is added to both of the latent features. Then, the latent features are sent to the UNet to predict the latent depth features $\hat{z}_d$, and the final depth predictions $\hat{d}$ are obtained via the decoder of the VAE. Note that both U-Net $\mathcal{U}$ and the VAE decoder $\mathcal{D}$ are trained to allow predictions in any range. Given the depth labels $d$ with its sparse mask $M$, the training loss is given by:

$$\mathcal{L}_{\text{depth}} = \mathbb{E}_{\mathbf{x},\hat{\mathbf{c}} \sim p_{\text{data}}} \|M \odot [\mathcal{D}(\mathcal{U}(\mathbf{z}^x, \hat{\mathbf{z}}^c)) - d]\| \tag{5}$$

**Sparse-View 3D Reconstruction** & **Pose Estimation.** Capturing sparse-view images with varying camera settings, particularly focal lengths, complicates object reconstruction using structure-from-motion (SfM) methods like COLMAP (Schönberger & Frahm, 2016) due to missing intrinsic parameters and low image overlap. While approaches like DUST3R (Wang et al., 2024) optimize both intrinsic and extrinsic parameters for reconstruction from sparse viewpoints, they struggle with significantly different intrinsic settings. To address this, we incorporate our estimated intrinsics as a geometry cue for the subsequent reconstruction in Wang et al. (2024), fixing the focal length during optimization to demonstrate the robustness of our approach. In Wang et al. (2024), the pointmap $X \in \mathbb{R}^{H \times W \times 3}$ is predicted, and the relative pose $P^* = [R^*|t^*]$ can be recovered via Procrustes alignment (Luo & Hancock, 1999). More details are provided in our appendix.

## 4 EXPERIMENTS

### 4.1 EXPERIMENTAL SETUP

**Datasets.** For camera intrinsic estimation, the training data is sourced from a variety of datasets, including NuScenes (Caesar et al., 2020), KITTI (Geiger et al., 2012), CityScapes (Cordts et al., 2016), NYUv2 (Nathan Silberman & Fergus, 2012), SUN3D (Xiao et al., 2013), ARKitScenes (Baruch et al., 2021), Objectron (Ahmadyan et al., 2021), MVImgNet (Yu et al., 2023), Hypersim (Roberts et al., 2021), Virtual KITTI (Cabon et al., 2020), Taskonomy (Zamir et al., 2018), and TartanAir (Wang et al., 2020). We adopt Waymo (Sun et al., 2020a), RGBD (Sturm et al., 2012), ScanNet (Dai et al., 2017), MVS (Fuhrmann et al., 2014), and Scenes11 (Chang et al., 2015) datasets for zero-shot testing.

For metric depth training, we use Taskonomy (Zamir et al., 2018), Hypersim (Roberts et al., 2021), TartanAir (Wang et al., 2020), Virtual KITTI (Cabon et al., 2020), Waymo (Sun et al., 2020b) and Argoverse2 (Wilson et al., 2021). Additionally, we incorporate 10k synthetic city samples collected by ourselves. The evaluation is performed on NuScenes (Caesar et al., 2020), ETH3D (Schöps et al., 2017), Diode (Vasiljevic et al., 2019), VOID (Wong et al., 2020), IBims-1 (Koch et al., 2020), NYUV2 (Nathan Silberman & Fergus, 2012). For more details, please refer to the appendix.

**Evaluation Protocols.** For camera intrinsic estimation, we follow the evaluation protocol of (Zhu et al., 2023; He et al., 2024) using the relative error:

$$e_f = \max\left(\frac{|f'_x - f_x|}{f_x}, \frac{|f'_y - f_y|}{f_y}\right), e_b = \max\left(2 \cdot \frac{|c'_x - c_x|}{w}, 2 \cdot \frac{|c'_y - c_y|}{h}\right) \quad (6)$$

For depth estimation, we use the Absolute mean relative error (A.Rel), the percentage of inlier pixels $\delta_i$ with threshold $1.25^i$ and scale-invariant error in log scale $\text{SI}_{\log} = 100\sqrt{\text{Var}\left(\varepsilon_{\log}\right)}$.

**Baselines.** For camera calibration, we compare our method with three non-diffusion based methods, Jin et al. (2023), Zhu et al. (2023) Piccinelli et al. (2024) and Veicht et al. (2024), one diffusion-based method (He et al., 2024). For metric depth estimation, we compare our method with 4 state-of-the-art methods. For additional reference, we also evaluate the generated depth using affine-invariant depth protocols with several affine-invariant depth depth estimation methods.

**Implementation Details.** Our models are built on the pre-trained Stable Diffusion V2.1 model (Rombach et al., 2022). To train camera intrinsic estimation model, we employ the AdamW optimizer with a learning rate of $3e^{-5}$ and train the model for 30,000 iterations with a total batch size of 196 on a cluster of 8 Nvidia A800 GPUs. For metric depth estimation, we use the same optimizer and learning rate with a total batch size of 96, and the training process takes approximately 5 days to converge. For all of our downstream 3D vision tasks, we did not use the ground truth camera image but instead relied on intrinsic parameters predicted by our diffusion model.

### 4.2 CAMERA INTRINSIC EVALUATION

We first present our monocular camera calibration results on five zero-shot datasets in Tab. 1. As shown, our method achieves the highest calibration accuracy. Compared to the concurrent work DiffCalib (He et al., 2024), our approach performs better due to the superior suitability of the proposed Camera Image for diffusion priors, allowing seamless integration with stable diffusion models. Among the methods, Unidepth (Piccinelli et al., 2024) shows the weakest performance, particularly in real-world, unconstrained scenarios such as the MVS dataset. This subpar result may stem from unbalanced training between the tasks of metric depth estimation and camera intrinsic estimation. Geometry-inspired methods (Jin et al., 2023; Veicht et al., 2024) face challenges in achieving strong performance on these datasets as they heavily rely on geometric information, such as vanishing points. Notably, our method also performs well on the highly challenging Scenes11 dataset (Chang et al., 2015), which features randomly shaped, moving objects, further demonstrating its robustness in extreme conditions.

### 4.3 DEPTH EVALUATION

**Metric Depth Comparison.** We evaluate the performance of our method on metric depth estimation task. The quantitative and qualitative results are presented in Tab. 2 and Fig. 6 respectively. Our work

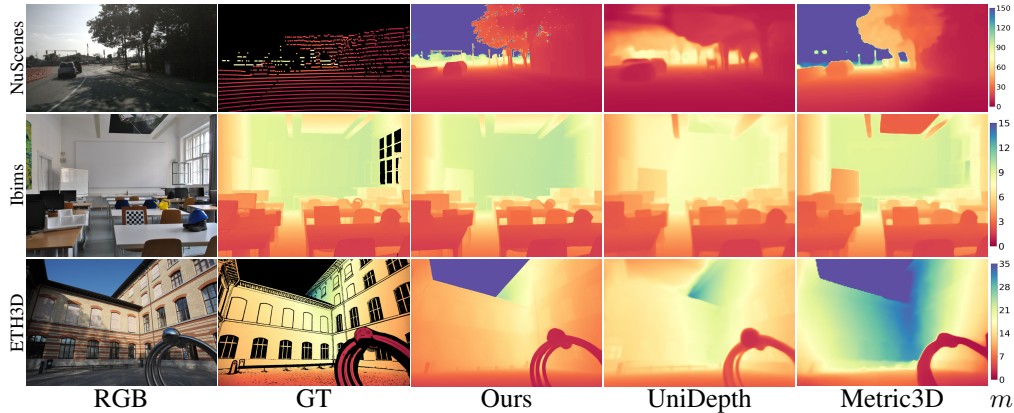

Figure 6: **Zero-Shot Metric Depth Estimation Results.** We present the predicted metric depth in both various scenes. Our method provides more detailed results and recovers accurate metric.

Table 2: **Comparison on Zero-Shot Metric Depth Evaluation.** We achieve comparable precision to state-of-the-art models while utilizing less training data.

| Method | NYU-V2 $\delta_1 \uparrow$ | $SI_{log} \downarrow$ | NuScenes $\delta_1 \uparrow$ | $SI_{log} \downarrow$ | ETH3D $\delta_1 \uparrow$ | $SI_{log} \downarrow$ | DIODE (Indoor) $\delta_1 \uparrow$ | $SI_{log} \downarrow$ | VOID $\delta_1 \uparrow$ | $SI_{log} \downarrow$ | IBims-1 $\delta_1 \uparrow$ | $SI_{log} \downarrow$ |
|---|---|---|---|---|---|---|---|---|---|---|---|---|
| iDisc | 93.8 | 8.85 | 39.4 | 37.1 | 35.6 | 27.5 | 23.8 | 15.8 | 55.3 | 20.3 | 48.9 | 13.2 |
| ZoeDepth | 90.1 | – | 28.3 | 31.5 | 35.0 | 17.6 | 36.9 | 12.8 | 63.4 | 15.9 | 58.0 | 10.9 |
| Metric3D | 92.6 | 9.13 | 72.3 | 29.0 | 45.6 | 18.9 | 39.2 | 11.1 | 65.9 | 16.2 | 79.7 | 10.1 |
| UniDepth | **97.2** | **6.41** | 83.3 | 22.9 | 22.9 | 13.1 | **60.4** | **9.01** | **88.5** | **8.26** | 79.4 | 8.88 |
| Ours | 93.1 | 8.35 | **85.7** | **19.8** | **49.0** | **9.08** | 42.2 | 13.3 | 73.1 | 15.3 | **88.5** | **8.27** |

Table 3: **Quantitative Comparison on 5 Zero-shot Affine-invariant Depth Benchmarks.** Despite targeting metric depth, we achieve performance comparable to SoTA affine-invariant depth methods.

| Method | NYUv2 AbsRel $\downarrow$ | $\delta1 \uparrow$ | KITTI AbsRel $\downarrow$ | $\delta1 \uparrow$ | ETH3D AbsRel $\downarrow$ | $\delta1 \uparrow$ | ScanNet AbsRel $\downarrow$ | $\delta1 \uparrow$ | DIODE-Full AbsRel $\downarrow$ | $\delta1 \uparrow$ |
|---|---|---|---|---|---|---|---|---|---|---|
| DiverseDepth (Yin et al., 2020) | 11.7 | 87.5 | 19.0 | 70.4 | 22.8 | 69.4 | 10.9 | 88.2 | 37.6 | 63.1 |
| MiDaS (Ranftl et al., 2022) | 11.1 | 88.5 | 23.6 | 63.0 | 18.4 | 75.2 | 12.1 | 84.6 | 33.2 | 71.5 |
| LeReS (Yin et al., 2021) | 9.0 | 91.6 | 14.9 | 78.4 | 17.1 | 77.7 | 9.1 | 91.7 | 27.1 | 76.6 |
| Omnidata v2 (Kar et al., 2022) | 7.4 | 94.5 | 14.9 | 83.5 | 16.6 | 77.8 | 7.5 | 93.6 | 33.9 | 74.2 |
| HDN (Zhang et al., 2022) | 6.9 | 94.8 | 11.5 | 86.7 | 12.1 | 83.3 | 8.0 | 93.9 | 24.6 | 78.0 |
| DPT (Ranftl et al., 2021) | 9.8 | 90.3 | 10.0 | 90.1 | 7.8 | 94.6 | 8.2 | 93.4 | 18.2 | 75.8 |
| Metric3D (Yin et al., 2023) | 5.8 | 96.3 | **5.8** | **97.0** | 6.6 | 96.0 | 7.4 | 94.1 | 22.4 | 78.5 |
| DepthAnything (Yang et al., 2024) | **4.3** | **98.1** | 7.6 | 94.7 | 12.7 | 88.2 | **4.2** | **98.0** | 27.7 | 75.9 |
| Marigold (Ke et al., 2024) | 5.5 | 96.4 | 9.9 | 91.6 | 6.5 | 96.0 | 6.4 | 95.1 | 30.8 | 77.3 |
| GeoWizard (Fu et al., 2024) | 5.2 | 96.6 | 9.7 | 92.1 | **6.4** | **96.1** | 6.1 | 95.3 | 29.7 | 79.2 |
| Ours | 4.8 | 97.1 | 8.5 | 93.5 | 7.1 | 95.3 | 5.7 | 96.5 | 25.6 | **79.4** |

significantly outperforms strong baselines such as Metric3D (Yin et al., 2023) by a large margin, and achieves comparable performance with a concurrent work Unidepth (Piccinelli et al., 2024). Based on the visualization results in Fig. 6, compared to UniDepth and Metric3D, our method presents sharper details and more accurate structural relationships for the captured scenes.

**Affine-invariant Depth Comparison.** Though our method is trained for metric depth, we transform the predicted depth into affine-invariant depth for broader comparisons. As shown in Tab. 3, our model achieves performance comparable to methods specifically designed for affine-invariant depth, such as Marigold (Ke et al., 2024) and GeoWizard (Fu et al., 2024), despite being designed for metric depth. As show in Fig. 7, we provide visual results in both in-the-wild and synthetic scenarios. Our approach consistently demonstrates superior spatial structural understanding, such as accurately distinguishing the tree in the background or the Pisa tower, which is correctly inferred to be closer than the nearby church.

## 4.4 MORE 3D VISION TASKS

**Monocular 3D Metrology.** We evaluate the accuracy of our camera intrinsic estimation and metric depth prediction by estimating the true size of objects captured by cameras with varying focal lengths. For example, using the car shown in Fig. 8, we estimate the distance between the wheels. Compared to Metric3D(Yin et al., 2023), our method provides more accurate distance estimates across different

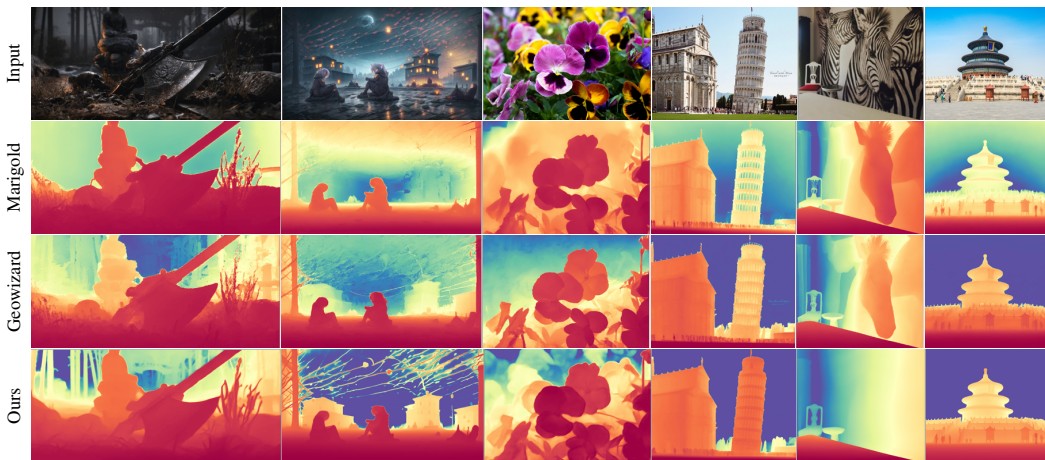

Figure 7: **Zero-shot qualitative affine-invariant depth results.** Our method demonstrates superior foreground-background differentiation (e.g., flower) and improved understanding (e.g., wall painting).

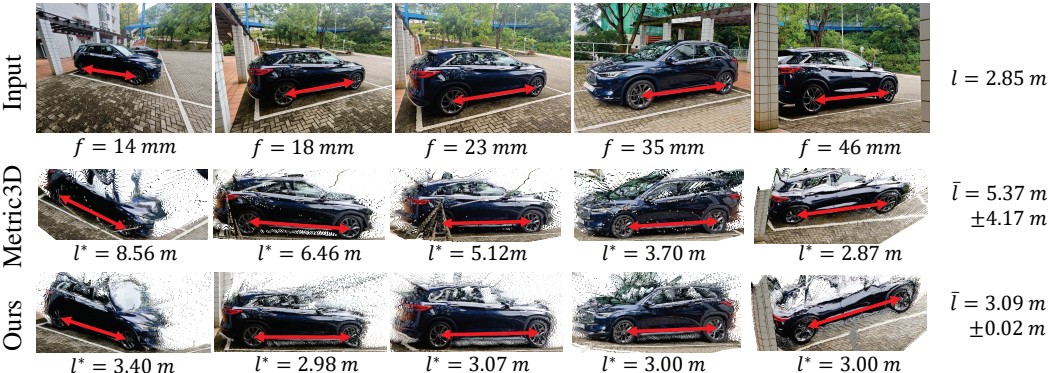

Figure 8: **Metrology of in-the-wild scenes.** Our method accurately recovers real-world metrics and demonstrates robustness to variations in focal length.

focal lengths and demonstrates robustness in both outdoor and indoor scenarios (see Fig. 11. However, performance decreases slightly with ultra-wide angles due to the scarcity of small focal length images in our training data, which could be improved by using more balanced datasets.

**Sparse-view 3D Reconstruction** & **Pose Estimation.** We further demonstrate that our estimated camera intrinsics can be effectively applied to sparse-view 3D reconstruction, especially when photos are taken with varying focal lengths. We evaluate the reconstruction results of Wang et al. (2024) with and without our estimated intrinsics. As shown in Fig. 9 and Fig. 13, reconstructions without intrinsic cues exhibit notable distortions and misalignments, whereas incorporating intrinsic cues significantly improves accuracy and alignment. Furthermore, as shown in Tab. 4, the reconstruction loss, represented by the mean relative distance between corresponding points, was reduced by around 20% on four in-the-wild scenes, confirming the effectiveness of using intrinsic cues for enhancing reconstruction quality. Additionally, as indicated in Tab. 5, pose estimation performance is also improved when intrinsic cues are used. Detailed experimental settings are provided in the appendix.

Table 4: **Relative distance error.** We compare the reconstruction performance with and without intrinsic cues.

|  | Sofa | Car | Pavilion | StoneWall |
|---|---|---|---|---|
| *w/o.* cue | 1.67 | 0.87 | 1.03 | 1.43 |
| *w.* cue | 1.37 | 0.68 | 0.68 | 1.06 |

Table 5: **Pose error.** We compare the pose error with and without intrinsic cue.

|  | $t_{rel}$ (m) | $r_{rel}$ (°) |
|---|---|---|
| *w/o.* cue | 1.17 | 5.02 |
| *w.* cue | 0.63 | 2.30 |

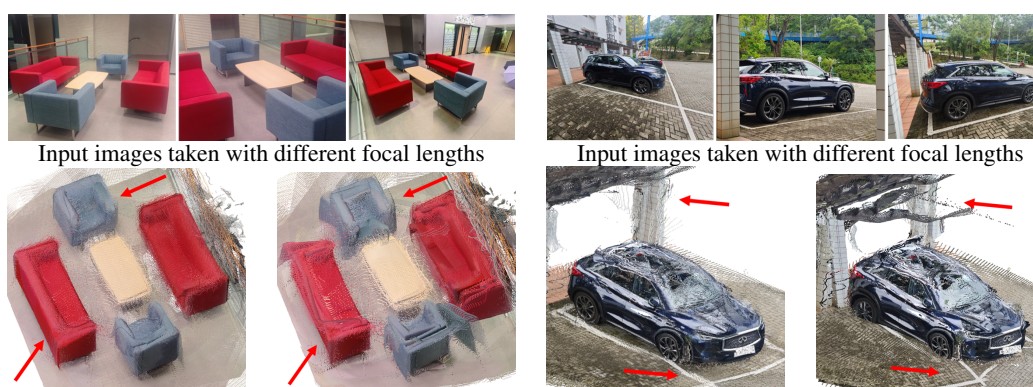

Figure 9: **Sparse View 3D Reconstruction with Intrinsic Cues.** We captured images with various focal lengths and present the reconstruction results. With intrinsic cues, our method achieves more accurate and better-aligned reconstructions.

### 4.5 ABLATION STUDY

**Ablation Study on Camera Calibration.** We evaluate the effectiveness of our proposed camera image representation and multi-resolution noise strategy through an ablation study on the GSV dataset (Anguelov et al., 2010), which includes 20 different camera intrinsic parameters. The study, conducted over 15k iterations, focuses on field-of-view prediction errors. As shown in Tab. 6, the naive approach using $c = [\theta, \phi, \theta]$ yields the poorest results due to the domain gap between the generated images and those produced by diffusion models. While multi-resolution noise(Kasiopy, 2023) improves performance slightly, it remains suboptimal. Incorporating our camera image representation significantly reduces the error, and the combination of both strategies produces the best results, demonstrating their complementary effectiveness.

**Ablation on Metric Depth Estimation.** We investigate the impact of our strategy on metric depth estimation by training a subset of our dataset for 20k iterations, with results shown in Tab. 7. Training solely on synthetic data has proven problematic, especially in indoor scenarios. Our second model, using a traditional multi-step denoising pipeline, integrates both virtual dense and sparse depth data but results in suboptimal performance due to the pipeline's inability to effectively recognize sparse areas in the ground truth. Additionally, prior methods that froze the VAE decoder during one-step training have shown to be inadequate for metric depth estimation, as demonstrated in our experiments. Omitting the camera image representation also slightly reduces accuracy. Interestingly, we found that the network can still produce relatively satisfactory results without explicit camera intrinsic guidance, contradicting previous studies that highlight the necessity of such information for metric depth estimation (Yin et al., 2023). We attribute this to the powerful pretrained SD model's capacity to capture subtle variations in camera intrinsic parameters.

Table 6: Ablation on Intrinsic Estimation.

| Multi-Res. Noise | Camera Image | Mean Error (°)↓ | Median Error (°)↓ |
|---|---|---|---|
| ✗ | ✗ | 24.36 | 25.74 |
| ✓ | ✗ | 9.10 | 7.00 |
| ✗ | ✓ | 6.72 | 6.33 |
| ✓ | ✓ | **2.54** | **2.01** |

Table 7: Ablation on Metric Depth Estimation.

| Ablation | NYU-v2 | | | KITTI | | |
|---|---|---|---|---|---|---|
| | $\delta_1 \uparrow$ | $SI_{log} \downarrow$ | A.Rel $\downarrow$ | $\delta_1 \uparrow$ | $SI_{log} \downarrow$ | A.Rel $\downarrow$ |
| Full Model | **85.8** | 8.17 | **13.5** | **89.1** | 13.3 | **11.7** |
| w.o Real data | 26.5 | 8.80 | 39.8 | 69.6 | 19.0 | 24.7 |
| w.o One step | 77.1 | 11.9 | 17.1 | 76.7 | 17.1 | 18.9 |
| w.o Decoder training | 76.7 | 11.1 | 48.0 | 83.5 | 14.5 | 13.4 |
| w.o Camera Image | 83.8 | **8.0** | 14.6 | 87.8 | **12.5** | 12.1 |

## 5 CONCLUSION

In conclusion, we introduced *DM-Calib*, a diffusion-based framework leveraging our proposed Camera Image for monocular camera calibration. By utilizing the strong priors of stable diffusion models, *DM-Calib* effectively estimates camera intrinsic parameters. Extensive experiments demonstrate its superior performance across a range of 3D vision tasks, consistently outperforming baseline methods in real-world scenarios and varied imaging conditions. Future work could address ultra-wide-angle images by incorporating more diverse training data and improve inference efficiency by developing a few-step diffusion (Luo et al., 2023) model to further enhance 3D vision tasks.

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

## A  MORE IMPLEMENTATION DETAILS

### A.1  CAMERA INTRINSIC PREDICTION

We train our model on a diverse range of datasets, ensuring balance by selecting one dataset per batch with equal probability and sampling from it. Most datasets follow the setup of Zhu et al. (2023), with additional data incorporated to better leverage the capabilities of stable diffusion. A detailed description of the datasets is provided in Tab. 9. Notably, our training set includes more data compared to He et al. (2024). For a fair comparison, we also report our results using the same training dataset and results is shown in Tab. 8. Regarding the Camera Image, we normalize its values to the range $[-1, 1]$ by dividing by $\pi$, and instead of force-resizing, we pad the Camera Image to a resolution of $768 \times 768$. Unlike previous works (Zhu et al., 2023; He et al., 2024) that directly resize images to a fixed size, we resize the images while preserving their aspect ratios, padding the remaining areas with zeros. This approach is necessary because the data we used were collected with various aspect ratios even within a single dataset. Following the data augmentation strategy applied in (Zhu et al., 2023), we randomly scale images up to twice their original size and then crop them back to the original resolution, with the camera intrinsics adjusted accordingly.

Table 8: **Monocular Camera Calibration on Zero-Shot Datasets.** We report the calibration errors for both focal length and optical center. *Small* means we train our model with same dataset with Zhu et al. (2023) and He et al. (2024).

| Method | Waymo | | RGBD | | ScanNet | | MVS | | Scenes11 | | Average | |
|---|---|---|---|---|---|---|---|---|---|---|---|---|
| | $e_f$ | $e_b$ | $e_f$ | $e_b$ | $e_f$ | $e_b$ | $e_f$ | $e_b$ | $e_f$ | $e_b$ | $e_f$ | $e_b$ |
| Ours-small | 0.138 | 0.033 | 0.051 | 0.012 | 0.084 | 0.023 | 0.080 | 0.010 | 0.071 | 0.014 | 0.085 | 0.017 |
| Ours | 0.115 | 0.036 | 0.041 | 0.010 | 0.089 | 0.024 | 0.087 | 0.008 | 0.061 | 0.010 | 0.078 | 0.017 |

### A.2  METRIC DEPTH PREDICTION

For metric depth prediction, we do not pad the images. Instead, we resize the maximum dimension of the images to 768 while maintaining their aspect ratios. Additionally, we apply random horizontal flipping and random cropping to enhance dataset diversity even in one dataset. Inspired by (Fu et al., 2024), we incorporate a "scene distribution decoupler" into our model through text-guided conditioned depth generation. Specifically, we utilize the CLIP tokenizer and encoder to encode the terms "indoor geometry" and "outdoor geometry" for different environments. Based on this setting, we treat the metric depth with different scale factor for indoor and outdoor: $s = \{s_{\text{in}}, s_{\text{out}}\}$, and the depth label become $d_s = d/s_i$ with $s_i \in s$ to fit the output of the training VAE decoder.

### A.3  MORE IMPLEMENTATION DETAILS AND DISCUSSIONS RELATING FIGURES AND TABLES.

**Fig. 2:** Our Camera Image is image-dependent, unlike other camera representations that are not. For other methods, lines can be plotted directly based on different FoV values. In contrast, we generate the line chart for the Camera Image using the GSV dataset (Anguelov et al., 2010), which includes 20 different types of cameras.

**Fig. 9** & **Fig. 13**: We take 20 to 25 images with five different focal lengths (same image focal lengths as shown in Fig. 8) and perform the reconstruction based on these images. Surrouding are cropped for better visulization. Our method complements sparse-view reconstruction methods like Dust3r (Wang et al., 2024) by providing intrinsic information, rather than serving as a direct comparison. Dust3r (Wang et al., 2024) delivers less accurate intrinsic estimation because it focuses on sparse-view reconstruction by generating point clouds for image pairs and performing global alignment to jointly optimize intrinsic calibrations and poses. This process is less robust and often converges to a local minimum. In contrast, our method is specifically designed to recover camera intrinsics. The results demonstrate that Dust3r achieves more accurate reconstruction when equipped with our estimated intrinsics.

**Tab. 5:** The pose estimation is compared against pseudo-ground truth generated using COLMAP (Schönberger & Frahm, 2016) from 60 images of a single object, leveraging the ground truth focal length for improved accuracy. For the reconstruction, we select 20 of these images and compare the pose estimation with and without intrinsic cues. Note that SE(3) and scale alignment are applied for the comparison.

Table 9: **Datasets List for camera calibration.** List of the training and testing datasets: number of images, scene type, and method of calibration. SfM: Structure-from-Motion.

| | Dataset | Images | Scene | Intrinsic |
|---|---|---|---|---|
| **Training Set** | NuScenes (Caesar et al., 2020) | 28k | Outdoor | Calibrated |
| | KITTI (Cordts et al., 2016) | 18 k | Outdoor | Calibrated |
| | CityScapes (Cordts et al., 2016) | 23k | Outdoor | Calibrated |
| | NYUv2 (Nathan Silberman & Fergus, 2012) | 6k | Indoor | Calibrated |
| | SUN3D (Xiao et al., 2013) | 33k | Indoor | Calibrated |
| | ARKitScenes (Baruch et al., 2021) | 48k | Indoor | Calibrated |
| | Objectron (Ahmadyan et al., 2021) | 33k | Indoor | SfM |
| | MVImgNet (Yu et al., 2023) | 27k | Indoor | SfM |
| | Hypersim (Roberts et al., 2021) | 54k | Indoor | Synthetic |
| | Virtual KITTI (Cabon et al., 2020) | 20k | Outdoor | Synthetic |
| | Taskonomy (Zamir et al., 2018) | 420k | Indoor | Rendered |
| | TartanAir (Wang et al., 2020) | 305k | Mix | Synthetic |
| **Testing Set** | Waymo (Sun et al., 2020a) | 800 | Outdoor | Calibrated |
| | RGBD (Sturm et al., 2012) | 160 | Indoor | Pre-defined |
| | ScanNet (Dai et al., 2017), | 800 | Indoor | Calibrated |
| | MVS (Fuhrmann et al., 2014) | 132 | Outdoor | Pre-defined |
| | Scenes11 (Chang et al., 2015) | 256 | Mixed | Pre-defined |

Table 10: **Datasets List for Metric Depth estimation.** List of the training and testing datasets for metric depth estimation: number of images, scene type, and method of Acquisition.

| | Dataset | Images | Scene | Acquisition |
|---|---|---|---|---|
| **Training Set** | Hypersim (Roberts et al., 2021) | 54k | Indoor | Synthetic |
| | Virtual KITTI (Cabon et al., 2020) | 20k | Outdoor | Synthetic |
| | Taskonomy (Zamir et al., 2018) | 40M | Indoor | RGB-D |
| | TartanAir (Wang et al., 2020) | 305k | Mix | Synthetic |
| | Argoverse2 Wilson et al. (2021) | 403k | Outdoor | LiDAR |
| | Waymo Sun et al. (2020b) | 223k | Outdoor | LiDAR |
| | Self-rendered | 10k | Outdoor | Synthetic |
| **Testing Set** | Diode Vasiljevic et al. (2019) | 771 | Mix | LiDAR |
| | ETH3D Schöps et al. (2017) | 454 | Outdoor | RGB-D |
| | IBims-1 Koch et al. (2020) | 100 | Indoor | RGB-D |
| | NuScenes Caesar et al. (2020) | 3k | Outdoor | LiDAR |
| | NYU Nathan Silberman & Fergus (2012) | 654 | Indoor | RGB-D |
| | VOID Wong et al. (2020) | 800 | Indoor | RGB-D |

**Tab. 3:** We assess the generalization ability across five zero-shot datasets by aligning the predicted depth $\hat{d}$ to the ground-truth depth $d$ with a scale factor $s$ and translation $t$, resulting in the aligned depth map $a = s \times \hat{d} + t$

**Fig. 8 and Fig. 11:** From a single input image, we first estimate the camera intrinsics and metric depth map, transform them into a 3D point cloud using the pinhole camera model, and calculate the 3D distance between key points.

**Procrustes alignment:** When pointcloud $X$ is given, the relative pose can be obtained by Procrustes alignment (Luo & Hancock, 1999):

$$R^*, t^* = \arg\min_{\sigma, R, t} \sum_i \left\| \sigma(RX_i^{1,1} + t) - X_i^{1,2} \right\|^2,$$

where $X^{1,2}$ represents the pointmap of image 1 in the coordinate frame of image 2. Then, a global alignment of the pointmaps is performed to further refine the pose and obtain the final aligned pointcloud reconstruction

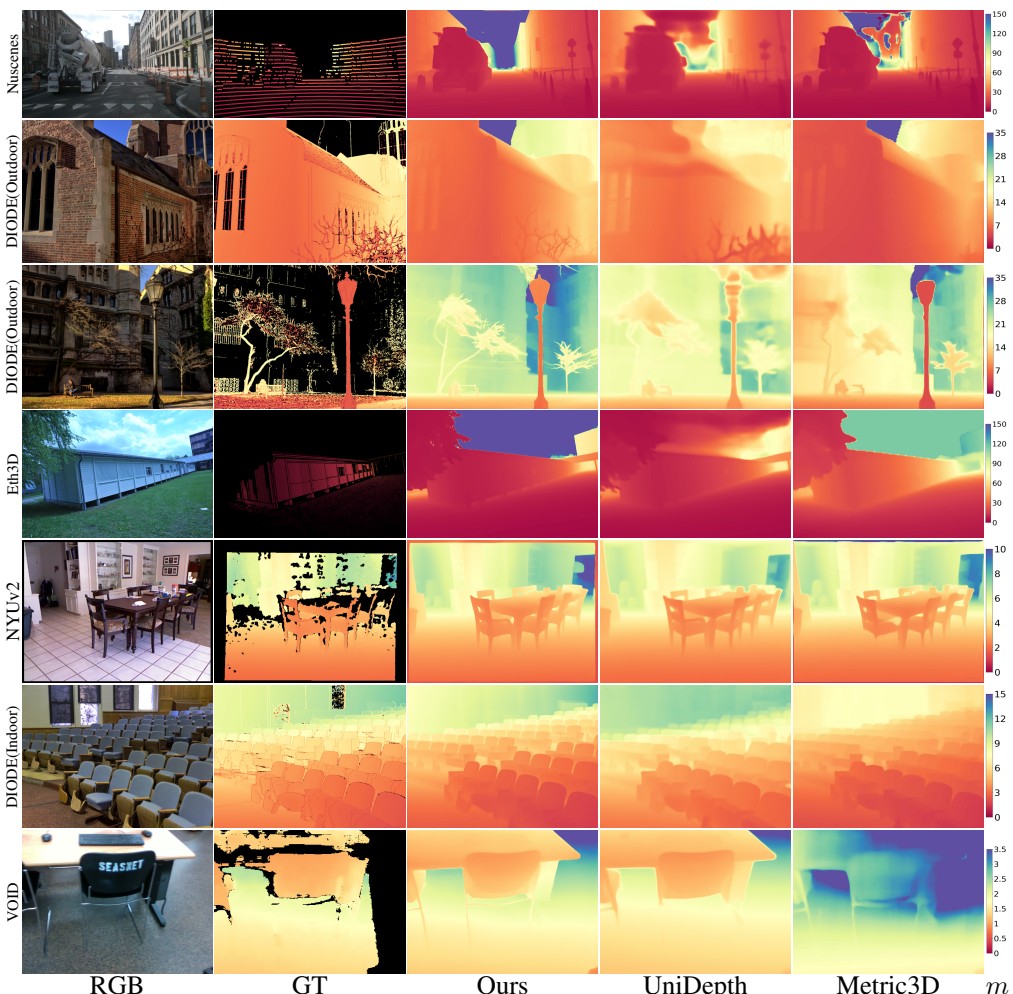

Figure 10: **Zero-Shot Metric Depth Estimation Results.** We present the predicted metric depth in both outdoor and indoor scenes. Our method provides more detailed results and recovers accurate metric depths.

# B   MORE EXPERIMENTAL RESULTS

## B.1   METRIC DEPTH

We show more qualitative metric depth prediction in Fig. 10.

## B.2   METROLOGIE

We show more Metrologie results in Fig. 11 compared with Metric3D (Yin et al., 2023).

We also present the metrologie results for UniDepth (Piccinelli et al., 2024) in Fig. 12. While it shows some limitations in focal estimation, this leads to slightly less accurate visualizations.

## B.3   3D RECONSTRUCTION

We show more qualitative 3D reconstruction results in Fig. 13.

## B.4   MESH RECONSTRUCTION

By using our predict metric depth, we can deduce corresponding normal map, and mesh can be reconstructed via the depth and normal map using BiNI algorithm (Cao et al., 2022). We present the

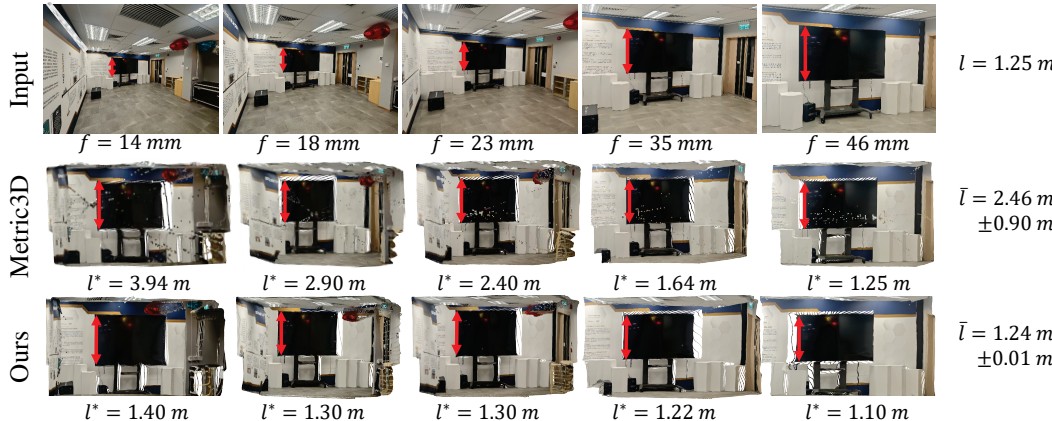

Figure 11: **Metrology of in-the-wild scenes.** Our method accurately recovers real-world metrics while demonstrating robustness to variations in focal length.

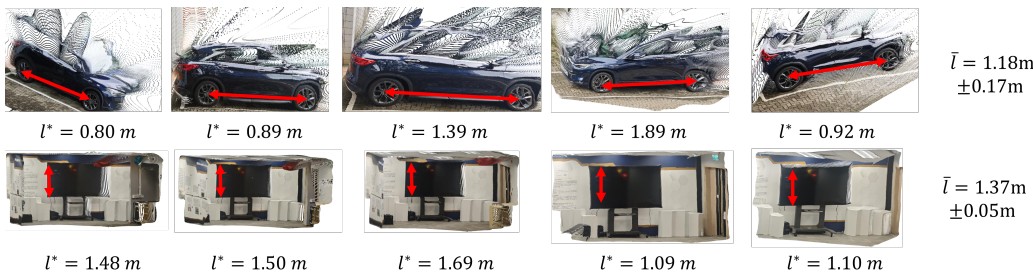

Figure 12: **Metrology of in-the-wild scenes for UniDepth.**

reconstruction result of Pisa tower in Fig. 7, and we show the reconstructed mesh in Fig. 14. Noting that we crop all background for better visualization.

### B.5 SINGLE VIEW 3D RECONSTUCTION

In this section, we present single-view 3D reconstruction of different camera focal length results using our estimated camera intrinsics and metric depth map. By applying the pinhole camera model, we transform the estimated intrinsics and depth map into a 3D point cloud. We demonstrate the robustness of our intrinsic estimation and depth prediction through in-the-wild single-view 3D reconstructions. Qualitative results can be found in Fig 15.

### B.6 THE IMPORTANCE OF PRINCIPAL POINT EVALUATION AND THE ASSESSMENT OF BOTH VERTICAL AND HORIZONTAL FOCAL LENGTHS

In our work, we evaluate the focal length as well as the principle points. Some previous works (Jin et al., 2023; Veicht et al., 2025) focuses solely on focal length. We prove the indispensability to evaluate the principal points. We have a significant amount of data where the principal point does not lie at the image center in certain datasets, and our model effectively learns the position of the principal points rather than ignoring them. To validate this, we conduct an ablation study comparing the error when assuming the principal point lies at the image center ($e_b$) with the error of our estimated principal point ($\hat{e}_b$). We show the results on Tab. 11.

Furthermore, not all datasets have $f_x = f_y$ (e.g., CityScapes dataset (Cordts et al., 2016) with $f_x = 2268.36$ and $f_y = 2225.54$). And our method is inherently capable of solving for both $f_x$ and $f_y$ and we take this into account to ensure more robust estimation and support future broader applications and datasets such as Diode (Vasiljevic et al., 2019).

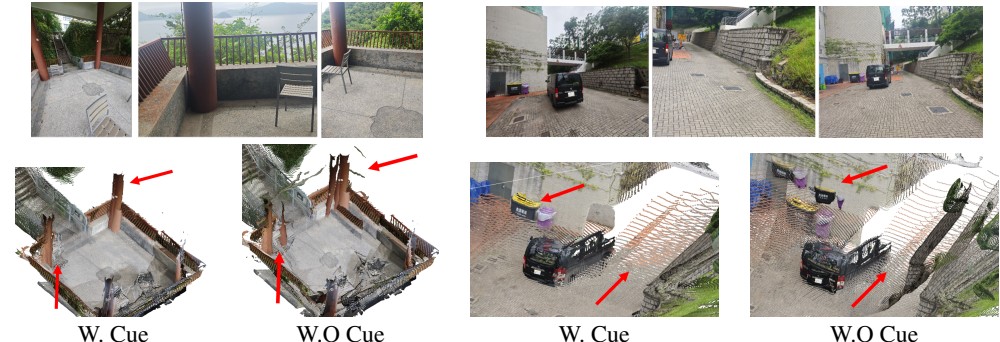

W. Cue         W.O Cue         W. Cue         W.O Cue

Figure 13: **Sparse view 3D reconstruction with intrinsic cue.** We captured images at different focal lengths and present the reconstruction results. With intrinsic cues, the reconstruction is more accurate and better aligned.

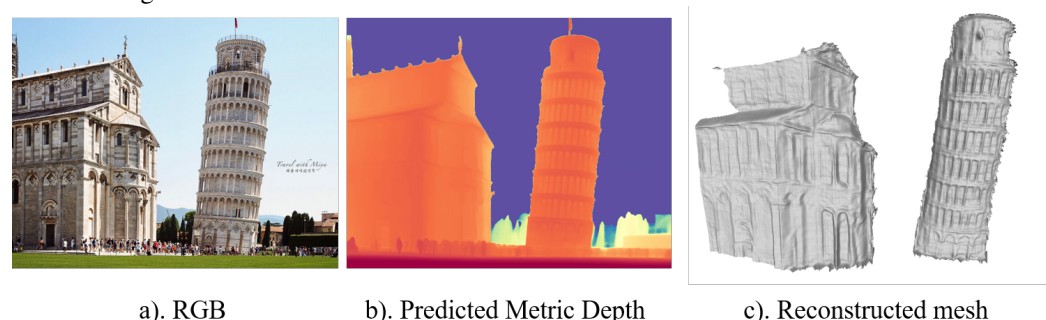

a). RGB         b). Predicted Metric Depth         c). Reconstructed mesh

Figure 14: **The reconstructed mesh using our predicted intrinsics and metric depth.**

### B.7   THE IMPORTANCE OF CAMERA IMAGE IN METRIC DEPTH ESTIMATION.

The camera image (intrinsic information) is essential for robust and accurate metric depth estimation. We present the $\delta_1$ results on three additional datasets in Tab.12, complementing the findings in Tab.7.

As shown, the absence of the camera image leads to a significant performance drop.

### B.8   TEST-TIME ENSEMBLING

To reduce the stochasticity of the process, we aggregate five predicted camera images by taking their mean. This significantly minimizes the randomness of the diffusion model, as evidenced by the small standard deviation in Tab. 13.

Without the aggregation, the standard deviation is sometimes not negligible, as presented in Tab. 14.

Table 11: **Principal points error** We compare the error of principle point estimation when assuming principal point lies at the image center with the error of our estimated principal point.

|  | NuScenes | KITTI | CityScapes | NYUv2 |
|---|---|---|---|---|
| $e_b$ | 0.051 | 0.021 | 0.055 | 0.050 |
| $\hat{e}_b$ | 0.007 | 0.014 | 0.011 | 0.009 |

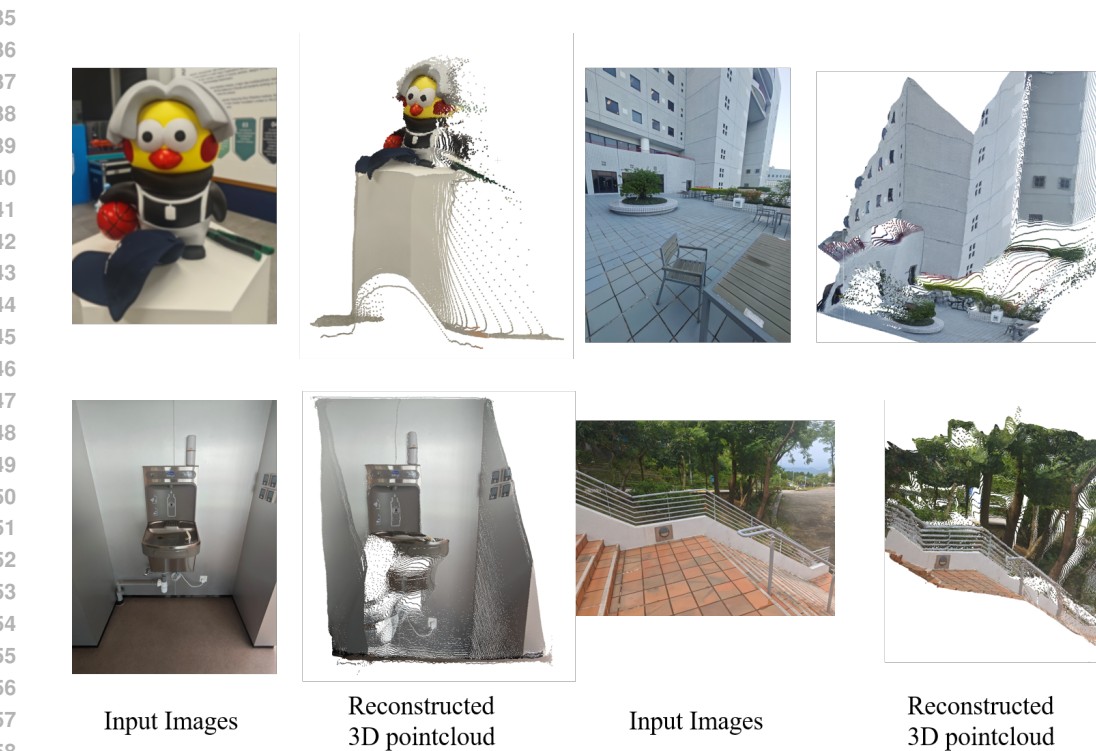

|  Input Images | Reconstructed 3D pointcloud | Input Images | Reconstructed 3D pointcloud |

Figure 15: **The reconstructed pointcloud from images with different camera focal length using our predicted instrinc and metric depth.**

Table 12: Ablation study on the effectiveness of camera images for metric depth estimation.

|  | ibims | Diode (indoor) | Diode (outdoor) |
| --- | --- | --- | --- |
| *w.* camera image | 88.7 | 50.1 | 41.0 |
| *w.o* camera image | 82.6 | 35.0 | 25.2 |

Table 13: **Standard Deviation of estimated intrinsics after ensembling.**

|  | Waymo | RGBD | ScanNet | MVS | Scenes11 | Average |
| --- | --- | --- | --- | --- | --- | --- |
| $e_f$ | $0.115 \pm 0.008$ | $0.041 \pm 0.002$ | $0.089 \pm 0.002$ | $0.087 \pm 0.006$ | $0.061 \pm 0.006$ | $0.078 \pm 0.006$ |
| $e_b$ | $0.036 \pm 0.001$ | $0.010 \pm 0.000$ | $0.024 \pm 0.000$ | $0.008 \pm 0.000$ | $0.010 \pm 0.001$ | $0.017 \pm 0.001$ |

Table 14: **Standard Deviation of estimated intrinsics after ensembling.**

|  | Waymo | RGBD | ScanNet | MVS | Scenes11 | Average |
| --- | --- | --- | --- | --- | --- | --- |
| $e_f$ | $0.115 \pm 0.035$ | $0.041 \pm 0.010$ | $0.089 \pm 0.024$ | $0.087 \pm 0.008$ | $0.061 \pm 0.009$ | $0.078 \pm 0.017$ |
| $e_b$ | $0.036 \pm 0.012$ | $0.010 \pm 0.001$ | $0.024 \pm 0.001$ | $0.008 \pm 0.001$ | $0.010 \pm 0.001$ | $0.017 \pm 0.001$ |

