# OpenReview forum: "Boost 3D Reconstruction using Diffusion-based Intrinsic Estimation"
_ICLR.cc/2025/Conference — Submitted to ICLR 2025_

### Official Review · Reviewer_DFVs · 2024-10-31

**Soundness:** 2
**Presentation:** 3
**Contribution:** 2
**Rating:** 6
**Confidence:** 4

**Summary:**

This paper presents DM-Calib, a diffusion-based approach that utilizes a unique Camera Image representation to estimate monocular camera intrinsics. The authors leverage a pre-trained stable diffusion model, adapting it to predict camera parameters from single images. The proposed method is evaluated on a range of datasets (KITTI, NuScenes, etc.) and is intended to support 3D vision tasks such as depth estimation, 3D reconstruction, and pose estimation.

**Strengths:**

1. The idea of embedding camera parameters as a “Camera Image” and feeding it into a diffusion model is innovative. I think this approach is a step forward in making monocular camera calibration feasible in single-image scenarios.

2 . The paper’s evaluations span multiple vision tasks, indicating an attempt to generalize across applications. Evaluating on KITTI, NuScenes, and other datasets gives a broad base for understanding DM-Calib’s performance.

**Weaknesses:**

From my point of view, the weakness mainly comes from the experiments part:

(1) I noticed that the baseline comparisons are missing some key controls. For example, it would be essential to include a version of DM-Calib with ablated components (e.g., without Camera Image or grayscale encoding) to clearly understand each component's contribution. The current results leave questions about the true utility of the Camera Image itself and whether it alone brings significant performance gains.

(2)  In Table 3, I noticed that while DM-Calib reports a slightly lower error than one baseline, the gains are minimal and within the standard deviation range. The small margin makes it hard to justify DM-Calib's superiority, and without significance testing, these small gains may not be meaningful.

(3) The ablation study provided in Table 4 is sparse and does not explain how each component of DM-Calib contributes to the final accuracy. I think that a more thorough examination of components, such as the role of grayscale encoding, is necessary. In particular, removing grayscale information and testing only the Camera Image format would have strengthened the argument.

**Questions:**

(1) Could the authors provide a more detailed breakdown of how much the Camera Image impacts the results independently from the rest of the model? Given its centrality, I would expect this to be a core component of the results.

---

> ### Author Response · Authors · 2024-11-18
>
> Thank you very much for reviewing our submission. We have addressed all concerns expressed in your reviews.
>
> - **Comment 1**: The reviewer finds the ablation study in Tab. 4 insufficient, as it lacks a detailed analysis of each DM-Calib component’s contribution to accuracy.
>
>     **Answer**: We would like to clarify that Table 4 is not the ablation study. Table 4 highlights the advantages of our proposed DM-calib in downstream applications like sparse-view 3D reconstruction. The ablation studies are actually presented in Tab. 6 and Tab. 7, where Tab. 6 analyze each DM-Calib component’s contribution to accuracy.
>
>     &nbsp;
>
> - **Comment 2**: The reviewer notes that DM-Calib shows minimal improvement over the baseline in Table 3 (comparison of affine-invariant depth estimation), with gains falling within the standard deviation range.
>
>
>     **Answer**:  Our primary contribution lies in the calibration of pinhole cameras, achieving state-of-the-art performance as shown in Table 1. We leverage the camera image for metric depth estimation (Table 2) and perform scale-shift alignment to derive affine-invariant depth, as most monocular depth estimation methods focus on this setting (Table 3). Notably, recovering metric depth is significantly more challenging than affine-invariant depth, yet our approach delivers competitive and comparable performance without sophisticated designs.
>
>     &nbsp;
>
> - **Comment 3**: The reviewer points out that baseline comparisons lack an ablation study of DM-Calib (e.g., without Camera Image or grayscale encoding) to clarify each component’s impact.
>
>     **Answer**: We conducted the mentioned ablation study in Tab. 6. We also shown this table below, demonstrating the effectiveness of the proposed camera image.
>     | Multi-Res     |   | Camera Image  |   | Mean Error ($^{\circ}$ )$\downarrow$  |   | Median Error ($^{\circ}$) $\downarrow$ |
>     |:-------------:|:---:|:-------------:|:---:|:-------------------------------------:|:---:|:--------------------------------------:|
>     | $\times$      |   | $\times$      |   | 24.36                                 |   | 25.74                                  |
>     | $\checkmark$  |   | $\times$      |   | 9.10                                  |   | 7.00                                   |
>     | $\times$      |   | $\checkmark$  |   | 6.72                                  |   | 6.33                                   |
>     | $\checkmark$  |   | $\checkmark$  |   | **2.54**                                  |   | **2.01**                                   |

---

> > ### Comment · Reviewer_DFVs · 2024-11-22
> >
> > Thank you for your detailed responses to the comments.
> >
> > Your rebuttal effectively addresses the concerns raised in the review and clarifies the distinct contributions of DM-Calib. The additional context provided for Tables 4, 6, and 7 is appreciated and improves the transparency of your analysis. I recommend incorporating these clarifications directly into the manuscript to ensure future readers can more easily follow the distinctions between evaluations and better understand the system’s contributions.
> >
> > Based on the additional clarifications and the thoughtful responses provided, I will raise my score to 6. Thank you for your detailed efforts in addressing the feedback.

---

> > > ### Author Response · Authors · 2024-11-23
> > >
> > > Thank you for reviewing our responses, updating the score, and providing valuable feedback; we will incorporate these clarifications into the manuscript.

---

### Official Review · Reviewer_wN9M · 2024-11-04

**Soundness:** 3
**Presentation:** 3
**Contribution:** 2
**Rating:** 6
**Confidence:** 4

**Summary:**

The paper repurposes Stable Diffusion (SD) for intrinsic and metric depth prediction. It proposes an image representation termed "Camera Image," which is a 3-channel image: the first two channels are related with ray directions, and the third channel is the grayscale image. The fine-tuned SD 2.1 can predict the corresponding Camera Image from the input image. Then, the intrinsics can be solved by RANSAC. The paper also showcases applications in metric depth estimation and sparse-view reconstruction (based on Dust3r). Experiments are conducted on various datasets. It outperforms baselines on camera calibration and achieves comparable results on depth estimation.

**Strengths:**

1. The proposed diffusion-based method for intrinsics prediction is simple but effective. It outperforms a recent SOTA baseline DiffCalib.

2. Compared to DiffCalib, the paper uses the grayscale image as the third channel, which makes it closer to the pretrained prior (Table 6). This makes training easier.

3. The paper conducts a good ablation study and many experiments on metric depth prediction and 3D reconstruction.

**Weaknesses:**

1. The overall idea is very similar to DiffCalib, which was released earlier this year. DiffCalib also fine-tunes Stable Diffusion 2.1 to estimate both the incident map for intrinsics and the depth map.

2. If the VAE is fine-tuned or simply rescaled to match the distribution before the diffusion model training, will the gray image still be necessary?

3. The diffusion model is stochastic. The standard deviation in the experiments should also be reported.

**Questions:**

L228: Explaining the ray vector r at its first appearance would benefit novice readers.

Typos:

L292: deonise → denoise

L293: encoder → encode

---

> ### Author Response · Authors · 2024-11-18
>
> Thank you for your insightful summary and for acknowledging our contribution. We provide detailed responses to all concerns in the following sections of this document.
>
> - **Comment 1**: The reviewer notes that the proposed method closely resembles DiffCalib, which also fine-tunes Stable Diffusion 2.1 to estimate both the incident map for intrinsics and the depth map.
>
>     **Answer**: To the best of our knowledge, DiffCalib was posted on arXiv several months ago and may not have undergone peer review. Furthermore, we believe that this work and ours were developed concurrently, and our method differs significantly from DiffCalib in two key aspects:
>     - Our proposed camera image provides a superior formulation for intrinsic estimation compared to the incident map introduced by DiffCalib, as demonstrated in Fig. 2 and Tab. 1. Below, we present the average errors for focal length ($e_f$) and principal point ($e_b$) for both methods (lower is better).
>
>         |           |   | $e_f$ |   | $e_b$ |
>         |:---------:|:---:|:-----:|:---:|:-----:|
>         | DiffCalib |   | 0.122 |   | 0.030 |
>         | Ours      |   | **0.078** |   | **0.017** |
>
>     - We estimate metric depth through a one-step deterministic process, unlike DiffCalib's diffusion-based affine-invariant depth. Notably, the estimated metric depth and intrinsics enable point cloud reconstruction, which is not directly achievable with affine-invariant depth [1].
>
>     &nbsp;
>
>
>     [1] Yin, Wei, et al. "Learning to recover 3d scene shape from a single image." Proceedings of the IEEE/CVF Conference on Computer Vision and Pattern Recognition. 2021.
>
>     &nbsp;
>
> - **Comment 2**: If the VAE is fine-tuned or simply rescaled to match the distribution before the diffusion model training, will the gray image still be necessary?
>
>
>     **Answer**:  We froze the VAE encoder and fine-tuned the decoder while training it concurrently with the U-Net for metric depth estimation, as shown in Fig. 5. The grayscale image remains necessary, as it is used to form the three-channel camera image. The VAE encoder is pre-trained on three-channel images and is only compatible with this format, which is why the grayscale image is reserved as part of the input.
>
>     &nbsp;
>
> - **Comment 3**: The diffusion model is stochastic. The standard deviation in the experiments should also be reported.
>
>     **Answer**: Thanks for point this out, we report our standard deviation as a complementary of our Tab. 1 below:
>     |       | Waymo           |   | RGBD            |   | ScanNet         |   | MVS             |   | Scenes11        |   | Average         |
>     |:-----:|:---------------:|:---:|:---------------:|:---:|:---------------:|:---:|:---------------:|:---:|:---------------:|:---:|:---------------:|
>     | $e_f$ | $0.115\pm0.008$ |   | $0.041\pm0.002$ |   | $0.089\pm0.002$ |   | $0.087\pm0.006$ |   | $0.061\pm0.006$ |   | $0.078\pm0.006$ |
>     | $e_b$ | $0.036\pm0.001$ |   | $0.010\pm0.000$ |   | $0.024\pm0.000$ |   | $0.008\pm0.000$ |   | $0.010\pm0.001$ |   | $0.017\pm0.001$ |
>
>
>
>     Specifically, to reduce the stochasticity of the process, we aggregate five predicted camera images by taking their mean. This significantly minimizes the randomness of the diffusion model, as evidenced by the small standard deviation in our results.
>
>     Without the aggregation, the standard deviation is sometimes not negligible, as presented below:
>
>
>     |       | Waymo           |   | RGBD            |   | ScanNet         |   | MVS             |   | Scenes11        |   | Average         |
>     |:-----:|:---------------:|:---:|:---------------:|:---:|:---------------:|:---:|:---------------:|:---:|:---------------:|:---:|:---------------:|
>     | $e_f$ | $0.115\pm0.035$ |   | $0.041\pm0.010$ |   | $0.089\pm0.024$ |   | $0.087\pm0.008$ |   | $0.061\pm0.009$ |   | $0.078\pm0.017$ |
>     | $e_b$ | $0.036\pm0.012$ |   | $0.010\pm0.001$ |   | $0.024\pm0.001$ |   | $0.008\pm0.001$ |   | $0.010\pm0.001$ |   | $0.017\pm0.001$ |
>
>
>     We will include this discussion in our revised version.
>
>     &nbsp;
>
> - **Comment 4**: Explaining the ray vector $\vec{r}$ at its first appearance would benefit novice readers.
>
>     **Answer**:Thanks for your advice. The ray vector $\vec{r} $  is the normalized camera ray originating from the camera center and passing through the pixel in the camera coordinate system.

---

> > ### Comment · Reviewer_wN9M · 2024-11-26
> >
> > Thanks for addressing my questions! I will keep the current rating.

---

> > > ### Author Response · Authors · 2024-11-27
> > >
> > > Thank you for your positive score and your valuable insights, which will undoubtedly help us refine and strengthen our work!

---

### Official Review · Reviewer_fdKi · 2024-11-04

**Soundness:** 3
**Presentation:** 2
**Contribution:** 3
**Rating:** 6
**Confidence:** 4

**Summary:**

In this work, authors propose DM-Calib, a monocular camera calibration method by fine-tuning a stable-diffusion model. The main contribution of the work is the image-representation of a pin-hole intrinsic matrix called "Camera Image." This representation allows authors to fine-tune a text-to-image stable-diffusion model as an image-to-camera-image diffusion model. From the generated camera image, authors estimate intrinsic using RANSAC. Authors show their superior camera calibration accuracy compared to non-diffusion and diffusion-based methods.

Authors show multiple applications of their camera calibration method: Specifically, they train Metric-depth estimation that uses a camera image + RGB image to predict one diffusion-step metric depth using a diffusion UNet. Authors also propose applications to sparse view 3D reconstruction, pose estimation, and 3D metrology, and show superior/comparable results to the SOTA methods in this field.

**Strengths:**

The proposed "Camera Image" shows easy integration with current diffusion-based generative models. Authors also show that intrinsics obtained using this method generalize in a zero-shot manner on multiple real-world indoor and outdoor datasets. This intrinsic representation, which closely resembles an RGB image, opens doors for other 3D reconstruction and calibration methods to be able to use large diffusion model priors.

Authors also show that they can train a single-step diffusion method that generates metric depth from input RGB + intrinsic camera image representation. This simple metric depth achieves comparable performance to specialized metric-depth estimation methods such as Metric3D, and also achieves comparable results to affine-invariant depth estimation methods.

**Weaknesses:**

While the proposed Camera Image representation for intrinsics is promising, authors did not discuss how to incorporate distortion into this image. Is the calibration limited to the Pinhole model? I would ask authors to clearly state in the abstract that this method can recover pinhole intrinsics from the image, without distortion parameters.

For the metric-depth estimation method, authors propose a single-step diffusion model. I see that it uses a single-step pass for diffusion, which essentially is just a forward pass to the UNet of Stable-diffusion. Is this a diffusion-based method? If yes, I would be happy if authors cite relevant sources.

One thing that is a bit unclear from the paper is that, does the metric-depth estimation model use "GT" camera-image, or a camera-image predicted from the diffusion model? If it uses "GT" camera-image, then the method is not comparable to other metric-depth estimation methods, many of which predict the method without intrinsics. I would request authors to explicitly mention in the paper if the camera-image used in metric depth estimation is GT or predicted one.

While I appreciate the many applications of Camera Intrinsic, describing so many applications is taking the focus away from the main theme of the paper, which is monocular camera calibration. I would propose authors to make the paper focused on camera calibration, such that readers can understand the main contribution of the paper easily. Some of the applications (Metrology, Pose estimation) can go in the appendix.

**Questions:**

- Is the input to the metric depth estimation module a "real" camera image? If yes, how can this method be compared to other methods?

- Why are the most recent approaches such as "GeoCalib" [1] excluded from comparison? I would be very interested in comparing this method against more "geometry"-inspired methods GeoCalib [1], that has also shown impressive results across many datasets.

References:

[1] Veicht, Alexander, et al. "GeoCalib: Learning Single-image Calibration with Geometric Optimization." arXiv preprint arXiv:2409.06704 (2024).

---

> ### Author Response · Authors · 2024-11-18
>
> Thank you very much for reviewing our submission. Your constructive suggestions and detailed comments have been invaluable in improving the quality of our work.
>
> - **Comment 1**: The reviewer appreciates the Camera Image representation but notes the lack of distortion handling, suggesting it's limited to the Pinhole model, and asks for clarification in the abstract about only recovering pinhole intrinsics.
>
>     **Answer**: Thank you for your comments and suggestions. We will clarify in our article that our focus is on recovering the Pinhole model from a single image without considering distortion parameters. Additionally, single-image undistortion is a well-studied problem with existing tools available for implementation [1, 2], which can be used as a preprocessing step for the input image of our method.
>
>   &nbsp;
>   &nbsp;
>
>     [1] Fan, Jinlong, et al. "Wide-angle image rectification: A survey." International Journal of Computer Vision 130.3 (2022): 747-776.
>
>     [2] cui,xingxing (2024). Single-Image-Undistort (https://github.com/cuixing158/Single-Image-Undistort), GitHub. Retrieved November 17, 2024.
>
>   &nbsp;
>
> - **Comment 2**: The reviewer questions whether the metric-depth estimation method, which uses a single-step pass through the UNet of Stable Diffusion, qualifies as a diffusion-based approach.
>
>
>     **Answer**:  Thank you for pointing that out. The one-step deterministic forward process proposed in our work is not a true diffusion-based method, as it does not involve a denoising process. Instead, we directly predict the target metric depth in a regression manner rather than noise in a diffusion scheme.
>
>   &nbsp;
>
> - **Comment 3**: The reviewer seeks clarification on whether the metric-depth estimation model uses a ''GT" camera image or a  one from our diffusion model DM-calib.
>
>     **Answer**: We did not use the ground truth camera image but instead relied on intrinsic parameters predicted by our diffusion model for all of our downstream 3D vision tasks, including Monocular Metric Depth Estimation, Sparse-View 3D Reconstruction \& Pose Estimation, Monocular 3D Metrology (discussed in the main article), as well as mesh and point-cloud reconstruction (presented in the appendix).
>
>   &nbsp;
>
> - **Comment 4**: Comparison with the recent methods like ''GeoCalib" [1].
>
>     **Answer**: Thank you for pointing that out. GeoCalib was released within a month of our submission, and we did not notice this approach in our evaluation. We have now compared our method with GeoCalib [1] as follows:
>
>     | Method   |   | Waymo |   | RGBD  |   | ScanNet |   | MVS   |   | Scenes11 |   | Average |
>     |----------|---|-------|---|-------|---|---------|---|-------|---|----------|---|---------|
>     | GeoCalib |   | 0.285 |   | 0.203 |   | 0.137   |   | 0.104 |   | 0.344    |   | 0.215   |
>     | Ours     |   | **0.115** |   | **0.041** |   | **0.089**   |   | **0.087** |   | **0.061**    |   | **0.078**  |
>
>     As we can see that our method outperforms GeoCalib [1] across all datasets. We will include these table in our revised version.
>
>   &nbsp;
>   &nbsp;
>
>     [1] Veicht, Alexander, et al. "GeoCalib: Learning Single-image Calibration with Geometric Optimization." arXiv preprint arXiv:2409.06704 (2024).
>
>   &nbsp;
>
> - **Comment 5**: The reviewer appreciates the applications of Camera Intrinsic but feels that describing too many. They suggest focusing on calibration in the main text and moving some applications to the appendix for better clarity.
>
>     **Answer**: Thank you for the suggestion. We will provide more detailed discussions on Camera Intrinsic and relocate some descriptions of downstream applications to the Appendix in our revised version.

---

> > ### Comment · Reviewer_fdKi · 2024-11-19
> > **Thank you for the rebuttal!**
> >
> > I thank authors for answering all my questions. Happy to see comparison with GeoCalib, would encourage authors to put this into appendix at least.
> >
> > The results are competitive to other Intrinsic estimation methods (Table 1), but still within variance threshold. Also, authors could not make concrete evidence of how this will improve metric depth estimation (since their ablation in table-7 shows even without camera image, they are able to achieve comparable accuracy). However, intrinsic formulation can be useful to other work. For this reason I am still keeping my score to 6.

---

> > > ### Author Response · Authors · 2024-11-20
> > >
> > > We appreciate your recognition of our work and your decision to maintain the score of 6. To address your concern about the effectiveness of the camera image, we conducted additional experiments:
> > >
> > > - The camera image (intrinsic information) is indispensable for robust and accurate metric estimation. We report results for $\delta 1$ on three additional datasets(higher is better):
> > >     | Doide (indoor)    |   | ibims |   | Doide (indoor) |   | Doide (outdoor) |
> > >     |:-----------------:|:---:|:-----:|:---:|:--------------:|:---:|:---------------:|
> > >     | w. camera image   |   | 88.7  |   | 50.1           |   | 41.0            |
> > >     | w.o. camera image |   | 82.6   |   | 35.0           |   | 25.2            |
> > >
> > >     As shown, the absence of the camera image leads to a significant performance drop, consistent with findings reported in [1, 2].
> > >
> > >
> > >     [1] Yin, Wei, et al. "Metric3d: Towards zero-shot metric 3d prediction from a single image." Proceedings of the IEEE/CVF International Conference on Computer Vision. 2023.
> > >
> > >     [2] Piccinelli, Luigi, et al. "UniDepth: Universal Monocular Metric Depth Estimation." Proceedings of the IEEE/CVF Conference on Computer Vision and Pattern Recognition. 2024.

---

### Official Review · Reviewer_G6iV · 2024-11-05

**Soundness:** 3
**Presentation:** 3
**Contribution:** 2
**Rating:** 6
**Confidence:** 4

**Summary:**

The paper introduces DM-Calib, a diffusion-based model for monocular camera calibration that leverages stable diffusion models trained on diverse data to estimate camera intrinsics from a single image. By integrating an image-based representation called the Camera Image, the method enables the extraction of camera parameters via a generative process with RANSAC for postprocessing. Demonstrated across various 3D vision tasks, DM-Calib outperforms existing methods, offering significant improvements in generalization and accuracy in camera calibration from monocular setups.

**Strengths:**

1. Compared to previous work that uses diffusion model for intrinsic prediction, this paper proposes a new spatial-aware representation for intrinsics, which is called Camera Image. The main motivation here is, to best leverage the power of pretrained diffusion model, the camera information should be incorporated as an image that fits the domain of the diffusion models. Therefore, this paper proposes to incorporate fx, fy, cx, and xy as the first two dimension, and uses the gray scale image as the third dimension. This design makes sense and seems to generally work well.

2. This paper did extensive experiments over different tasks and datasets, which effectively proves the ability of the proposed method.

**Weaknesses:**

1. The main concern comes from the Sparse-view 3D Reconstruction & Pose Estimation experiment. It seems, the authors claim that the proposed monocular view method can achieve better intrinsic estimation than mutli view method dust3r. Such a conclusion looks quite weird to the reviewer, especially given the fact that, although monocular depth estimation has achieved great success in the recent two years with the support of diffusion models, its accuracy is still not as good as those multi-view methods. The task of monocular metric depth estimation is very related to monocular intrinsic estimation, so the reviewer would expect there should not be such a huge difference. If so, does the result of "Sparse-view 3D Reconstruction & Pose Estimation experiment" imply that the intrinsic parameters estimated by dust3r are far away from accurate, or is there any specific setting? The authors mentioned that "Detailed experimental settings are provided in the appendix" in L480 but the reviewer did not find it.



2. The reviewer agrees that it is best to consider all of fx, fy, cx, and cy. However, looking at the datasets used for training, it seems most of them strictly have a principal point lying at the image center, e.g., all the self-driving datasets, almost all the synthetic datasets, and all the SfM-annotated (i.e., COLMAP) datasets. Furthermore, if I remember it correctly, most of these datasets have fx=fy. This leads to a question that, what is exactly learned by the model? Or in other words, if using a representation, just [fx, fy, gray] (with fx and fy normalized), will the performance be worse, or even better? It would be beneficial if the authors can provide more examples and analysis to prove that the parameters such as cx cy are not ignored by the model.

**Questions:**

The reviewer recognizes the extensive engineering effort evident in this paper and the promising performance of the implemented system. However, the underlying principle needs a further proof and a more thorough discussion of the sparse reconstruction experiments is needed. The primary concern is the Sparse-View 3D Reconstruction & Pose Estimation experiment, where the results appear unreasonable to the reviewer. Further clarification and justification of these outcomes are necessary.

---

> ### Author Response · Authors · 2024-11-18
>
> Thank you very much for reviewing our submission. Your constructive suggestions and comments have been invaluable in improving the quality of our work. We have addressed all concerns expressed in your reviews.
>
> - **Comment 1**: The reviewer finds it strange that the author's  monocular method achieves better intrinsic estimation than the multi-view method, Dust3r.
>
>     **Answer**: Thank you for your comment. Our method complements sparse-view reconstruction methods like Dust3r by providing intrinsic information, rather than serving as a direct comparison. Dust3r delivers less accurate intrinsic estimation because it focuses on sparse-view reconstruction by generating point clouds for image pairs and performing global alignment to jointly optimize intrinsic calibrations and poses. This process is less robust and often converges to a local minimum. In contrast, our method is specifically designed to recover camera intrinsics. The results demonstrate that Dust3r achieves more accurate reconstruction when equipped with our estimated intrinsics.
>
>   &nbsp;
>
> - **Comment 2**: The reviewer concern that most training datasets have a principal point at the image center and often $f_x = f_y$. They wonder whether a simpler representation [$f_x$, $f_y$, gray] might perform well or not, and request examples to demonstrate that $c_x$ and $c_y$ are not being ignored by the model.
>
>     **Answer**:  We have a significant amount of data where the principal point does not lie at the image center in certain datasets, and our model effectively learns the position of the principal points rather than ignoring them. To validate this, we conduct an ablation study comparing the error when assuming the principal point lies at the image center ($e_b$) with the error of our estimated principal point ($\hat{e}_b$). The results are presented below (lower is better):
>
>     |             | NuScenes  | KITTI | CityScapes | NYUv2 |
>     |:-----------:|:---------:|:-----:|:----------:|:-----:|
>     | $e_b$       | 0.051     | 0.021 | 0.055     | 0.050   |
>     | $\hat{e}_b$ | **0.007**     | **0.014** | **0.011**      | **0.009** |
>
>   &nbsp;
>
>     As we demonstrate the importance of accurately estimating the principal point, a formulation such as $[f_x, f_y, \text{gray}]$ is not reasonable.
>
>     Furthermore, not all datasets have $f_x = f_y$ (e.g., CityScapes dataset with $f_x = 2268.36$ and $f_y=2225.54$). And our method is inherently capable of solving for both $f_x$ and $f_y$ and we take this into account to ensure more robust estimation and support future broader applications and datasets such as Diode [1].
>
>   &nbsp;
>   &nbsp;
>
>     [1] Vasiljevic, Igor, et al. "Diode: A dense indoor and outdoor depth dataset." arXiv preprint arXiv:1908.00463 (2019).

---

> > ### Comment · Reviewer_G6iV · 2024-11-25
> >
> > Thanks for your response. Overall it makes sense to me now. Please make sure to clarify about the comparison to dust3r and add more details, as the current statement will look quite weird to those readers from the multi-view geometry community.
> >
> > I am glad to adjust my score correspondingly.

---

> > > ### Author Response · Authors · 2024-11-25
> > >
> > > Thank you for considering our rebuttal and updating the score! We really appreciate your recognition of our work and your feedback.

---

### Author Response · Authors · 2024-11-25

We sincerely thank the reviewers for their constructive comments and valuable suggestions. In response, we have conducted new analyses, revised the manuscript accordingly (the updated version has been uploaded, with changes highlighted in blue), and provided detailed responses to each reviewer's concerns.

Summary of important changes:
- Clarified in the manuscript that our focus is on recovering the Pinhole model from a single image, without accounting for distortion parameters.
- Added a comparison with the recent method "GeoCalib" [1].
- Clarified the comparison to Dust3r.
- Provided more detailed discussions on Camera Intrinsic and relocated some descriptions of downstream applications to the Appendix.
- Justified the evaluation of principal points (Tab. 11).
- Highlighted the importance of the camera image for metric depth estimation (Tab. 12).
- Reported the standard deviation of our intrinsic estimation (Tabs. 13 and 14).

[1] Veicht, Alexander, et al. "GeoCalib: Learning Single-image Calibration with Geometric Optimization." arXiv preprint arXiv:2409.06704 (2024).

---

### Meta-Review · Area_Chair_Sz41 · 2024-12-20

**Metareview:**

This paper tackles the task of predicting camera intrinsics given a single input image, and also shows downstream applications of this prediction improving tasks like metric depth estimation or sparse-view reconstruction. The key insight is that  intrinsics can be parametrized via a spatial camera image, and stable diffusion can be leveraged to conditionally predict this representation, which can in turn be used to condition downstream geometry prediction tasks. On the plus side, the approach of adapting stable diffusion to predict the “camera map” is novel, and the augmentation of the grayscale image to help make the camera image better aligned with SD domain is a neat idea. The empirical results also clearly demonstrate the efficacy of this representation, as well as the benefits of conditioning downstream tasks on predicted intrinsics.

There are some concerns though that the focus on the appliactions almost detracts from the central task of intrinsic prediction, and this could have been studied in more depth (including non-pinhole cameras). The more significant concern, however, is that the central insight regarding the parametrization is similar to one in prior works. Specifically, DiffCalib leveraged a similar representation, and although this work maybe concurrent, there are earlier instantiations of similar “camera images”. For example, this parametrization relates to “generalized cameras” long used in computer vision for calibration [A,B] (in fact, these are applicable beyond pinhole cameras) with the exact same idea of parametrizing intrinsics via per-pixel ray directions (in fact, these representations generalize it even further). This representation has also been incorporated in diffusion framework for predicting camera intrinsics and extrinsics [C]. Although the application and implementation this paper considers is different (and the concatenation of grayscale image is a neat addition), the insight about the representation cannot be claimed as novel.

While the reviewer ratings are borderline positive, none of the reviewers strongly advocate for this work. Combined with the concerns above, the AC feels this paper would benefit from a resubmission that better re-positions its contributions in context of the prior works (and perhaps tackles calibration in more challenging settings beyond a pinhole camera).

[A] A General Imaging Model and a Method for Finding its Parameters. *Grossberg and Nayar, ICCV 2001*
[B] Why Having 10,000 Parameters in Your Camera Model is Better Than Twelve. *Schopps et. al., CVPR 2019*
[C] Cameras as Rays: Pose Estimation via Ray Diffusion. *Zhang et. al., ICLR 2024*

**Additional Comments On Reviewer Discussion:**

The reviewers raised come concerns about the relation to other methods, applicability beyond pinhole cameras, and several aspects of implementation/comparisons. While the author response clarified the latter, the AC feels that the similarity of the camera image to prior (uncited) works is a concern, and that limitation to pinhole-only setting for a paper focused on calibration is a bit limiting.

---

### Decision · Program_Chairs · 2025-01-22

Reject